# The Odorant-Binding Proteins AspiOBP1 and AspiOBP2 in *Aleurocanthus spiniferus* Are Involved in the Perception of Host Volatiles

**DOI:** 10.3390/ijms26188784

**Published:** 2025-09-09

**Authors:** Zhifei Jia, Zeyu Qin, Xiaoyu Ge, Yongyu Xu, Zhenzhen Chen

**Affiliations:** State Key Laboratory of Wheat Improvement, College of Plant Protection, Shandong Agricultural University, Tai’an 271000, China; jiazf0525@163.com (Z.J.); 17606296383@163.com (Z.Q.); gexiaoyugongzuo@163.com (X.G.); xuyy@sdau.edu.cn (Y.X.)

**Keywords:** *Aleurocanthus spiniferus*, odorant binding proteins, volatile binding, host selection, RNAi pesticides

## Abstract

*Aleurocanthus spiniferus*, an invasive pest native to Southeast Asia, exhibits rapid dispersal capacity and high eradication resistance. In recent years, there have been continuous records of its invasion into new host plants. Odorant-binding proteins (OBPs) are essential at the peripheral level of olfaction, and their olfactory function has been partially confirmed by research. This study explores the functions of key OBPs mediating host selection by measuring the in vivo and in vitro binding capabilities of OBPs from *A. spiniferus* to host volatiles. Under exposure to more than five host volatiles, the two *OBPs*, *AspiOBP1* and *AspiOBP2*, exhibited significant differential transcriptional regulation. AspiOBP1 exhibited good binding affinity to (*Z*)-3-hexenol and 3-carene, and with binding energies greater than −3 kcal/mol, ARG-79 might be the critical amino acid site for AspiOBP1 binding to host volatiles. AspiOBP2 exhibited no binding to any of the six tested volatiles in fluorescent competitive binding assays. Adults fed with *dsAspiOBP1* showed significantly reduced behavioral and EAG responses to the attractant 3-carene and two repellents [(*Z*)-3-hexenol and nonanal]. Adults fed with *dsAspiOBP2* lost both behavioral and EAG responses to the attractant 3-carene and the repellent (*Z*)-3-hexenol. The findings of this study not only elucidate the binding mechanisms between OBPs of *A. spiniferus* and host volatiles but also provide new targets for the future development of novel plant-derived insecticides and RNA-based pesticides to control this pest.

## 1. Introduction

*Aleurocanthus spiniferus* (Quaintance) belongs to the order Hemiptera, family Aleyrodidae. Adults congregate and pierce-suck on tender leaves, while all larval stages except the first-instar larvae remain stationary on the undersides of leaves, feeding by piercing and sucking [1]. The honeydew excreted by this pest promotes the growth of sooty mold, which significantly reduces the photosynthetic efficiency of leaves [1]. These characteristics present significant control and management challenges. It has been recognized as an invasive pest native to Southeast Asia. Within a century, it had spread rapidly to Asia, Africa, the Americas, Australia, the Pacific Islands, and other regions [2,3,4,5]. The first detection of this species in the EPPO (European and Mediterranean Plant Protection Organization) regions occurred in 2008 in the Lecce district of the Apulia region, located in southeastern Italy, marking its initial reported presence in these areas [6]. Since then, the pest rapidly expanded to the southwestern regions of Italy and was also detected in the Balkan Peninsula, showing a trend of further northward invasion towards France [7,8]. It is considered one of the major threats to citrus production in Asia, Australasia, and the Nearctic zone [5,7,9]. Except for the main elective hosts *Citrus* spp. and Theaceae, *A. spiniferus* was recorded on several botanical families, e.g., Rosaceae, Leguminosae, Moraceae, Vitaceae, Punicaceae, Simaroubaceae, Ericaceae, Ranunculaceae, Anacardiaceae, and is still in the process of adapting to new hosts [1,7,10].

Odorant-binding proteins (OBPs) are small extracellular proteins whose main function is to capture odors and pheromones, and transport them to receptors [11,12]. Recently, olfactory-related proteins in *A. spiniferus* have been identified through transcriptomic analysis, with five OBPs being detected. These OBP genes are predominantly expressed in the head (including the antennae) [13]. OBPs are capable of recognizing a variety of volatile compounds [14,15] and act as chemosensory solubilizers, transporters, and ligand filters, mediating the activation of odorant receptors (ORs) [16]. Access to recombinant proteins has generated comprehensive data on OBPs’ binding affinities, molecular docking with bioactive compounds, and their well-characterized three-dimensional crystal structures [17,18,19]. Classic OBPs are defined by their structural feature of six *α*-helices linked by three disulfide bonds [20] and represent the most extensively studied and reviewed subfamily of OBPs to date [14,21,22,23,24]. The sequence motif patterns can vary significantly among different OBP subfamilies. For instance, the atypical OBP subfamily features additional cysteines in the C-terminal region [20], the minus-C OBP subfamily is characterized by only four cysteine residues [25,26], and the plus-C OBP subfamily contains three extra cysteines along with a conserved proline [21]. The diversity and non-homology of OBPs among insect genera can serve as an advantage for developing species-specific semiochemicals and even insecticides. This will further advance the use of OBPs as targets and tools for designing pest control agents.

Host volatiles play a crucial role as signaling chemicals in the interactions between insects and their host plants [27,28]. (*E*)-2-hexenal, linalool, nonanal, 3-carene, hexanol, and (*Z*)-3-hexenol have been confirmed to be present in the volatile compounds emitted by the leaves or fruits of the host plants of *A. spiniferus* [29,30,31,32,33,34,35]. Previous studies have demonstrated that (*E*)-2-hexenal, linalool, 3-carene, and hexanol act as attractants for *A. spiniferus*, while nonanal and (*Z*)-3-hexenol function as repellents [36]. These compounds influence the host-selection behavior of *A. spiniferus* across tea cultivars, identifying them as key chemicals for exploring this pest’s chemical ecology [37]. Nevertheless, the precise role of OBPs in *A. spiniferus* in detecting these volatile organic compounds remains mechanistically elusive.

In this study, volatile-induced expression alteration was used to preliminarily screen key AspiOBPs that may be involved in the perception of six volatiles. Fluorescence binding assay and molecular docking were used to validate the in vitro binding ability of key AspiOBPs with six ligands and to predict the key binding sites. Finally, we explored the role of AspiOBPs in sensing host volatiles. This study aims to provide deeper insights into the olfactory mechanisms whereby *A. spiniferus* identifies host volatiles, which will contribute to the advancement of new control strategies such as RNA pesticides and innovative synthetic compounds.

## 2. Results

### 2.1. Host Volatiles-Induced Changes in AspiOBPs Expression

When exposed to the attractant 3-carene, *AspiOBP3* was significantly upregulated, while *AspiOBP1*, *AspiOBP2*, and *AspiOBP7* were significantly downregulated (Figure 1a and Table A1). The attractant (*E*)-2-hexenal induced a notable upregulation of *AspiOBP1*, while *AspiOBP2*, *AspiOBP3*, and *AspiOBP5* were significantly downregulated (Figure 1b and Table A1). Exposure to the attractant linalool resulted in significant downregulation of *AspiOBP1* and *AspiOBP3* (Figure 1c and Table A1). The attractant hexanol induced a marked downregulation of *AspiOBP2* (Figure 1d and Table A1). *AspiOBP1*, *AspiOBP2*, and *AspiOBP5* were significantly downregulated under the influence of the repellent (*Z*)-3-hexenol (Figure 1e and Table A1). The repellent nonanal induced significant downregulation of *AspiOBP1*, *AspiOBP2*, *AspiOBP3*, and *AspiOBP5* (Figure 1f and Table A1). After treatment with more than five compounds, the relative expression levels of *AspiOBP1* and *AspiOBP2* underwent significant changes.

### 2.2. Fluorescence Binding Property Analysis

After removing the signal peptide and terminator, the recombinant AspiOBP1 and AspiOBP2 were successfully induced and expressed in *E. coli* (Figure 2). Their predicted molecular weights were approximately 16 and 27 kDa, respectively. However, due to the extremely low expression level of AspiOBP1, binding analysis could not be performed. Therefore, we only evaluated the binding affinity of recombinant AspiOBP2 with six host volatiles. The results showed that the Kd value of the AspiOBP2/bis-ANS complex was 2.26 ± 0.47 μmol/L (Figure 3). AspiOBP2 did not bind to any of the tested ligands (Figure 3).

### 2.3. Three-Dimensional Modeling and Molecular Docking

Due to the low expression level of AspiOBP1, it was not possible to evaluate its in vitro binding capacity using the fluorescence competitive binding assay. Therefore, this study conducted protein model prediction and molecular docking for AspiOBP1. Furthermore, this study did not obtain a qualified protein model for AspiOBP2. The protein model of AspiOBP1 was constructed using the SWISS-MODEL website (Figure 4), and its secondary structure was analyzed (Table A2). Molecular docking experiments were performed using Autodock 4.2.6 software to dock 3-carene, hexanol, (*E*)-2-hexenal, (*Z*)-3-hexenol, nonanal, and linalool with the three-dimensional model of AspiOBP1 (Figure 5). The docking energies between AspiOBP1 and each ligand were calculated (Table 1). The hydrophobic interactions and hydrogen bonds between AspiOBP1 and each ligand were analyzed using the “Protein-Ligand Interaction Profiler” website (Table A4 and Table A5).

The results showed that all docking binding energies were negative, with the binding energy between AspiOBP1 and (*Z*)-3-hexenol being −3.1 kcal/mol (Table 1), indicating the best binding characteristics. This was followed by 3-carene and (*E*)-2-hexenal, with binding energies of −3.07 and −2.47 kcal/mol, respectively (Table 1). The potential interacting residues of AspiOBP1 within a 4 Å range around the ligands were all hydrophobic residues, mainly including TYR-77, ARG-79, GLU-106, and MET-109 (Figure 5; Table A4). Additionally, ARG-79 formed hydrogen bonds with two attractants [hexanol, (*E*)-2-hexenal, and linalool] and one repellent [(*Z*)-3-hexenol], with bond lengths of 2.03, 1.85, 2.53, and 2.23 Å, respectively (Figure 5; Table A5).

### 2.4. RNAi Experiments of AspiOBP1 and AspiOBP2

As shown in Figure 6, compared to the control treated with distilled water (CK), the expression levels of *AspiOBP1* and *AspiOBP2* were significantly reduced by 53.82% and 90.97%, respectively, following treatment with *dsAspiOBP1* and *dsAspiOBP2*.

EAG tests revealed that, compared to the control (CK and *dsGFP*), adults fed with *dsAspiOBP1* exhibited significantly reduced EAG responses to 3-carene, nonanal, and (*Z*)-3-hexenol, while adults fed with *dsAspiOBP2* showed significantly reduced EAG responses to 3-carene and (*Z*)-3-hexenol. However, both the control and the *dsAspiOBPs*-treated treatments displayed low EAG responses to (*E*)-2-hexenal (Figure 7).

Y-tube olfactometer tests showed that, compared to the control (CK and *dsGFP*), adults fed with *dsAspiOBP1* exhibited reduced behavioral responses to two attractants [3-carene, (*E*)-2-hexenal] and two repellents [(*Z*)-3-hexenol, nonanal] (Figure 8). Adults fed with *dsAspiOBP2* lost behavioral responses to two attractants [3-carene, (*E*)-2-hexenal] and one repellent [(*Z*)-3-hexenol] (Figure 9).

## 3. Discussion

Recent studies have shown that exposure to odorants can lead to changes in the transcription of olfactory receptor genes associated with the reception of these odorants [38]. This technique is referred to as deorphanization of receptors based on expression alteration of mRNA levels (DREAM) [38]. Previous studies have utilized this method for the preliminary screening of key CSPs in *A. spiniferus* [36], as well as for the functional characterization of OBPs in other insects, such as HoblOBP and DcitOBP [39,40]. In this study, we analyzed the changes in the transcriptional levels of *AspiOBPs* induced by six host volatiles. The results revealed that the transcriptional levels of *AspiOBP1* and *AspiOBP2* were significantly up- or down-regulated after treatment with more than five compounds. Therefore, we focused on AspiOBP1 and AspiOBP2 to analyze their roles in the recognition of plant volatiles. Most importantly, changes in gene expression following exposure to significant plant volatiles can provide a pathway for identifying key plant volatiles that interact with crucial olfactory proteins among a vast array of plant volatiles [41].

Fluorescence competitive binding assays are commonly used to analyze the interactions between insect OBPs and ligands [22,42,43], and their reliability has been demonstrated through in vivo experiments [44,45]. Since *AspiOBP2* is known to be highly expressed in the heads of *A. spiniferus* adults [13], we employed the fluorescence competitive binding assay to analyze the binding properties of recombinant AspiOBP2 protein with host volatiles. Unfortunately, AspiOBP2 does not bind to any of the six host volatiles. However, when the expression of the *AspiOBP2* gene was knocked down, *A. spiniferus* adults lost their behavioral responses to two attractants [3-carene, (*E*)-2-hexenal] and one repellent [(*Z*)-3-hexenol]. In insects, various types of OBPs contribute to olfactory perception, often exhibiting combinatorial recognition through functional complementation or interaction [46,47]. Therefore, disrupting the expression of a single OBP may have complex effects on the expression of other OBPs. Nevertheless, fully understanding the intricate mechanisms of olfactory reception remains a significant challenge.

Some key residues located within the hydrophobic cavity are believed to facilitate the odorant-binding interactions between OBPs and their ligands [48,49]. Due to the inability to obtain highly expressed AspiOBP1 for in vitro binding assays, we utilized SWISS-MODEL to generate a structural model of AspiOBP1 and evaluated its binding energy with selected odorants. 3-Carene and (*Z*)-3-hexenol exhibited good binding affinities, with binding energies of −3.07 and −3.1 kcal/mol, respectively. 3-Carene was found to reside within a hydrophobic cavity formed by six amino acid residues (GLN-76, TYR-77, ARG-79, GLU-106, MET-109, TYR-117) without forming hydrogen bonds. This suggests that hydrophobic interactions alone can achieve good binding affinity. Hydrophobic interactions play a crucial role in mediating ligand binding, as their function is not constrained by the strict geometric requirements of intermolecular interactions such as hydrogen bonding [50,51,52]. Although the binding energy between AspiOBP1 and linalool is the lowest at −1.78 kcal/mol, it still forms hydrophobic interactions with two amino acid residues (GLU-80, GLU-106) and establishes hydrogen bonds with three amino acid residues (ASN-78, ARG-79, GLU-80). This demonstrates that the binding capability between proteins and ligands is influenced by multiple factors and is not solely dependent on the number of hydrophobic interactions or hydrogen bonds formed.

RNA interference (RNAi) triggered by dsRNA represents a groundbreaking technology for investigating the functions of specific target genes and is regarded as an effective tool for pest control [53]. Feeding is the most efficient method for delivering dsRNA into adult insects [40]. Although the transcription of *AspiOBP1* could not be completely silenced, partial knockdown significantly reduced the EAG and behavioral responses of *A. spiniferus* adults to certain tested volatiles [3-carene, nonanal, and (*Z*)-3-hexenol]. Similar results have also been observed in AgosOBP2, AlinOBP4, and DcitOBP7 [40,54,55].

Irrespective of OBP gene interference, the EAG responses to (*E*)-2-hexenal remained consistently low, aligning with the findings of our prior research [36]. Research has shown that not all insect OBP genes involved in olfactory functions are specifically expressed in the antennae [13]. Some OBPs are also widely expressed in non-olfactory tissues, such as mouthparts, legs, midguts, glands, and other non-olfactory tissues [56]. Similar to TcOBPC12, which exhibits high expression levels in the epidermis, this suggests that OBPs might play roles in additional physiological processes beyond olfactory detection [57]. The OBPs responsible for sensing (*E*)-2-hexenal could potentially be located in tissues other than the antennae, and further investigations, such as immunolocalization analyses, are necessary to confirm this hypothesis.

In the EAG tests, the antennal responses of *A. spiniferus* to each tested volatile were generally weak, with an average amplitude below 0.2 mV. Similar results have been observed in the responses of *Diaphorina citri* to host volatiles, which may be related to the relatively low number of olfactory receptors on their antennae. Only 11 receptors were found on each antenna of both female and male *D. citri* [40]. This conclusion, however, does not seem to apply to *A. spiniferus*, as its antennae possess a rich variety and quantity of sensilla [13]. An almost possible explanation is that the shorter antennae (Female: 296 ± 11 μm; Male: 247 ± 7 μm) of *A. spiniferus* limit their chances of getting good EAGs [13].

## 4. Materials and Methods

### 4.1. Insect Collection and Chemicals

Adult *A. spiniferus* specimens were collected from tea plants at Shandong Qianrun Ecological Agriculture Development Co., Ltd. in Tai’an, Shandong Province, China (32°08′ N, 117°43′ E). After collection, the adults were acclimated in environmental incubators set to 26 ± 2 °C, 70 ± 5% RH, and a 16 L: 8 D photoperiod for 24 h. During this period, approximately 90% of the adults were able to survive. Because the male sex ratio was low, preventing us from obtaining sufficient males for experiments, 1-day-old adult females and males were combined for testing. Hexane (analytical grade; Tianjin Kaitong Chemical Reagent Co. Ltd., Tianjin, China) served as the solvent. The following compounds were purchased from Macklin Inc. (Shanghai, China): 3-Carene (90%, CAS: 13466-78-9), (Z)-3-hexenol (98%, CAS: 928-96-1), hexanol (99%, CAS: 111-27-3), linalool (98%, CAS: 78-70-6), (E)-2-hexenal (98%, CAS: 6728-26-3), and nonanal (96%, CAS: 124-19-6).

### 4.2. Formatting of Mathematical Components

The volatile induction experiment was conducted following the methodology described by Jia et al. [36]. The tested compounds were prepared at a concentration of 100 μg/μL, with hexane used as the solvent. In a 20 mL glass vial, 100 adults were introduced. A filter paper strip (1 × 1 cm) soaked with 10 μL of the compound solution was placed inside the vial. The control group was treated with hexane alone. Both the control and experimental groups were maintained in an incubator under conditions of 26 ± 2 °C, 70 ± 5% relative humidity, and a photoperiod of 16 L: 8 D. After 4 h of exposure, the adults were transferred to cryotubes, flash-frozen in liquid nitrogen, and stored at −80 °C. Each treatment was replicated four times.

The relative transcript abundance of five AspiOBPs before and after compound induction was detected using the CFX96 Real-Time System (Bio-Rad, Hercules, CA, USA). The reaction mixture (20 μL) consisted of 10 μL of 2× ChamQ SYBR qPCR Master Mix (Vazyme, Nanjing, China), 1 μL each of forward and reverse primers, and 8 μL of cDNA (diluted 10-fold). A three-step PCR protocol was employed: initial denaturation at 94 °C for 30 s (1 cycle), followed by 40 cycles of amplification at 95 °C for 5 s and 60 °C for 30 s. The melting curve of the PCR products was measured under the conditions of 95 °C for 15 s, 60 °C for 1 min, and 95 °C for 15 s. Each sample was subjected to four technical replicates and four biological replicates. Gene expression levels were quantified using the 2^−ΔΔCt^ relative quantification method [58]. The relative expression levels after the control (CK) and compound treatments were analyzed using a two-tailed Student’s *t*-test.

### 4.3. Gene Cloning and Sequence Analysis

All gene sequences and expression data were obtained from the NCBI Sequence Read Archive (SRA) database under the accession number PRJNA792195. The signal peptide-coding sequences of AspiOBP genes were removed. Based on the open reading frame (ORF) sequences of the genes and the restriction sites of the expression vector pET-30a (+), protective base sequences and specific restriction enzyme recognition sequences were added to the 5′ end of the primers (Table A3). The recognition sequences for *EcoRI* and *BamHI* enzymes (Vazyme, Nanjing, China) were G^AATTC and G^GATCC, respectively.

Total RNA was extracted from adult *A. spiniferus* using a total RNA extraction kit (Vazyme, Nanjing, China). RNA integrity was verified by 1% agarose gel electrophoresis, and its purity was preliminarily assessed. The RNA concentration was measured using an Eppendorf BioPhotometer D30 (Eppendorf AG, Hamburg, Germany). An RNA sample was considered qualified if both the A260/A230 and A260/A280 ratios fell within the range of 1.8 to 2.2. cDNA was synthesized using a cDNA synthesis kit (Vazyme, Nanjing, China) in a total reaction volume of 20 μL. The target gene fragments were amplified from the cDNA. The 50 μL PCR reaction consisted of 25 μL of 2× Taq Plus Master Mix II, 1 μL each of forward and reverse primers, 5 μL of template DNA, and 18 μL of ddH_2_O. The PCR amplification program included an initial denaturation at 95 °C for 3 min, followed by 35 cycles of denaturation at 95 °C for 15 s, annealing at 60 °C for 20 s, and extension at 72 °C for 45 s, with a final extension at 72 °C for 10 min. The PCR products were purified and verified by 1% agarose gel electrophoresis. Qualified PCR products were stored at −20 °C for subsequent recombination experiments.

The pET-30a (+) vector was digested with the selected enzymes. The 50 μL digestion reaction consisted of 1 μg of plasmid, 5 μL of 10× rcutsmart Buffer, 1 μL each of *BamHI* and *EcoRI*, and ddH_2_O to make up the volume. The reaction was carried out at 37 °C for 90 min, followed by 65 °C for 20 min. The linearized vector obtained was used for recombination experiments. The target fragments were recombined with the vector using the Clon Express^®^ II One Step Cloning Kit (Vazyme, Nanjing, China). The 20 μL recombination reaction consisted of 4 μL of linearized vector, 4 μL of the target gene, 4 μL of Buffer, 2 μL of Exnase II, and 6 μL of ddH_2_O. The reaction was performed at 37 °C for 30 min. The recombinant plasmids were transformed into DH5*α* competent cells (Vazyme, Nanjing, China). Bacterial cultures with correct amino acid sequences were selected and added to Luria–Bertani (LB) liquid medium containing kanamycin (50 mg/L), followed by shaking at 37 °C for 15 h at 210 rpm. The bacterial culture (1 mL) was sent to BGI Genomics Co., Ltd. (Beijing, China) for sequencing verification. The sequencing results were compared against the ORF amino acid sequences on the official NCBI website. Finally, the recombinant plasmids containing the correct sequences were extracted from *Escherichia coli* and transformed into BL21 competent cells.

### 4.4. Expression and Purification of Recombinant Proteins

Positive clones were selected from the LB plate containing bacteria transformed with the recombinant plasmid. Due to the low protein expression level, the his-tag was not removed to ensure the smooth running of the binding experiment. Protein expression was induced by adding 1.0 mmol/L isopropyl-*β*-d-thiogalactoside (IPTG) and incubating at 37 °C for 8 h in LB medium. After induction, the bacterial culture was centrifuged at 4 °C and 5000 rpm for 10 min to collect the target-expressing bacterial cells. The proteins were analyzed using 15% SDS-PAGE and subsequently purified using affinity chromatography with a nickel column (GE Healthcare, Waukesha, WI, USA). The protein samples were eluted stepwise with 20 mmol/L, 50 mmol/L, 100 mmol/L, 250 mmol/L, and 500 mmol/L imidazole solutions. The elution fraction containing the highest concentration of the target protein with minimal impurities was selected for dialysis and ultrafiltration. The dialysis was carried out using a standard cellulose membrane (MWCO: 8–10 kDa) from Spectrum Labs, Inc. (Repligen Corporation, Waltham, MA, USA) against a 20 mmol Tris-HCl buffer (pH 8.0) at 4 °C for 12 h with three buffer changes. Ultrafiltration was subsequently performed using Amicon^®^ Ultra-15 centrifugal filters (MWCO: 10 kDa, Merck KGaA, Darmstadt, Germany) at 4000× *g* and 4 °C to concentrate the protein sample. The quality of the proteins was assessed using SDS-PAGE gel electrophoresis, and the protein concentration was determined using the BCA assay. The purified proteins were stored at −80 °C.

### 4.5. Fluorescence Competitive Binding Assay

The fluorescence competitive binding assays for six volatiles were conducted using a Cary Eclipse fluorescence spectrophotometer (Agilent Technologies, Palo Alto, CA, USA). 4,4′-Dianilino-1,1′-binaphthyl-5,5′-disulfonic acid, dipotassium salt (bis-ANS) was used as the fluorescent probe. The instrument parameters were set as follows: emission wavelength scanning range of 300–550 nm and excitation wavelength of 295 nm. The recombinant proteins were dissolved in 50 mmol/L Tris-HCl buffer (pH 7.4) at a final concentration of 2 µmol/L. The fluorescent probe and ligands were dissolved in chromatographically pure methanol at a final concentration of 1 mmol/L. The dissociation constant (Kd) of bis-ANS was determined using the Scatchard equation, and the binding affinity of the recombinant proteins to the six ligands was evaluated based on previous studies [59]. GraphPad Prism 9.0 was used to calculate and analyze the IC50 values (ligand concentration at 50% reduction in fluorescence intensity) and the inhibition constants (Ki) [60,61,62]. Each treatment was replicated three times.

### 4.6. Three-Dimensional Modeling and Molecular Docking

The amino acid sequences without signal peptides were subjected to BLAST sequence alignment analysis using the NCBI database https://blast.ncbi.nlm.nih.gov/Blast.cgi (accessed on 14 September 2024). Alignment results with E-values less than 10-5 and the smallest E-value (minimum value of 0) were saved. The template with the highest BLAST algorithm score was selected as the template for protein modeling. The target protein structure was predicted using the SWISS-MODEL server https://swissmodel.expasy.org/ (accessed on 14 September 2024) and optimized through a neural network-based method. The protein model with the highest score was chosen as the final structure of the target protein. The obtained protein models were analyzed using Procheck, Verify_3D, and ERRAT programs to evaluate their rationality (Figure A1 and Figure A2).

Molecular docking was performed following the method described by Wang et al. [63]. The protein models and ligands were docked using Autodock 4.2.6 software. The binding energy was calculated using the Lamarckian Genetic Algorithm (LGA). In the Autodock tools, the protein and ligand molecules were opened, and the docking box parameters were set. The docking box size was set to *x* = 40, *y* = 40, *z* = 40, with a grid point spacing of 0.375 Å, ensuring that the docking box covered the entire protein molecule. The center grid point coordinates were set to −1.986, 2.882, −0.029. The AutoGrid program was used to calculate the grid point energies, generating a series of Map files representing interatomic interactions. The AutoDock program was then run to evaluate and rank the ligands based on conformation and energy parameters. The docking results produced 50 best conformations, and the conformation with the lowest binding energy was selected for further analysis and evaluation. Images were generated and analyzed using PyMOL 3.9.0.

### 4.7. Synthesis of Double-Stranded RNA (dsRNA) and Treatment

RNAi experiments were performed following the experimental protocol of Jia et al. [36]. Double-stranded RNA (dsRNA) was synthesized using the T7 RiboMAX™ Express RNAi Kit (Promega, Madison, WI, USA) according to the manufacturer’s instructions. Specific primers containing T7 promoter sequences (Table A3) were designed for this purpose. The synthesized dsRNA was diluted to a concentration of 100 ng/μL using a 30% sucrose solution. Twenty adult *A. spiniferus*, starved for 1 h, were transferred into a food-grade plastic cup (upper diameter: 38 mm, lower diameter: 30 mm, height: 30 mm). The cup opening was sealed with a plastic film (Bemis, Inc., Neenah, WI, USA), and 20 droplets of dsRNA (1 μL each) were added to the film. A second layer of stretched film was then covered the first, and 20 needle holes were punctured through both layers. To prevent escape, the diameter of the needle holes was smaller than the cross-sectional width of the adults. Controls included dsRNA of green fluorescent protein (*dsGFP*, 436 bp) at the same concentration and distilled water (CK). Both treatments and controls were placed in an environmental incubator set at 26 ± 2 °C, 70 ± 5% relative humidity, and a photoperiod of 16L:8D. After 24 h of feeding, EAG and behavioral tests were conducted. Total RNA was extracted from the adults collected at the end of feeding treatments for qPCR analysis to evaluate RNA interference efficiency. Each treatment was replicated four times. Statistical differences in relative expression levels among *dsAspiOBP1*, *dsAspiOBP2*, *dsGFP*, and distilled water (CK) treatments were analyzed using one-way analysis of variance (ANOVA), with mean comparisons performed using Tukey’s-b multiple range test.

### 4.8. EAG Tests

The Electroantennographic (EAG) recordings of adult *A. spiniferus* to host volatiles were tested following the methods described by Gu et al. [64] and Tang et al. [65]. Six volatiles were diluted to concentrations of 100 μg/μL using hexane, with 10 μL solution dropped to filter paper strips (5 × 60 mm). The filter paper was quickly placed into a 14.5 cm Pasteur pipette and used as a stimulus source after 30 s. A control was prepared by dropping 10 μL of hexane to the filter paper. Timing began immediately after the antennae were excised from the insect, and EAG responses were recorded starting 5 min later. Each antenna was stimulated with three randomly selected volatiles from the test set, following the sequence: control, random volatiles, control, random volatiles, and control. Specifically, the tip of the Pasteur pipette was inserted into a steel tube through a small hole (approximately 11 cm from the tube opening) to a depth of about 3 mm. The stimulation duration was 0.3 s, with a 60-s interval between stimuli [66]. A total of 15 antennae were tested with only one antenna randomly selected per adult.

The EAG instrument (Syntech Ltd., Hilversum, The Netherlands) was connected to a computer via an IDAC-2 signal acquisition unit. Under a stereomicroscope, the reference and recording electrodes (Ag-AgCl) were connected to the base and tip of the antenna, respectively. The antenna was positioned approximately 2 mm from the opening of the steel tube. A 1 L/min airflow, purified through activated carbon and humidified, passed through the steel tube (14 cm long, 8 mm inner diameter) and carried the volatiles over the antenna. The testing conditions were maintained at 25 ± 2 °C and 46–78% relative humidity. The EAG responses were expressed as relative values, calculated by subtracting the average of the control (hexane) responses before and after each stimulus from the sample response. The resulting difference represented the relative EAG response to the sample stimulus. The experimental data were analyzed using one-way analysis of variance (ANOVA), with mean comparisons performed using Tukey’s-b multiple range test.

### 4.9. Two-Choice Olfactometry for dsRNA-Treated A. spiniferus

The Y-tube olfactometer test was adapted from the method described by Han and Chen [67]. The test compounds were prepared at a concentration of 100 μg/μL, with hexane used as the solvent. The Y-tube olfactometer consisted of a base and two arms, each 10 cm in length and 1 cm in internal diameter, with a 90° angle between the arms. In the odor source containers on both sides, 10 μL of the test volatile compound and hexane were placed, respectively. An electric vacuum pump was connected to the base of the Y-tube. The incoming air was filtered through activated charcoal, regulated by a flow meter, and humidified using water-saturated cotton balls. All components were connected using polytetrafluoroethylene (PTFE) tubing. The airflow through each arm was maintained at 100 mL/min. A total of 80 adults were tested individually. Each individual was given a choice between the volatile compound and hexane. The choice of adult was recorded after 10 min. After every 10 adults tested, the Y-tube was thoroughly cleaned in sequence using ethanol, acetone, and deionized water, and the positions of the two arms were swapped to avoid positional bias. The same testing procedure was repeated four times for each odor source treatment. After testing one odor source, the Y-tube, odor source containers, and other glass components are cleaned in sequence using ethanol, acetone, and deionized water, and dried in an oven at 120 °C before reuse. The activated charcoal in the filter was reactivated by heating in an oven at 100 °C for 4 h. After cooling, the activated charcoal was stored in a sealed glass bottle for future use. The bioassays were conducted between 9:30 and 15:30 in a laboratory maintained at 22–28 °C, 65–75% relative humidity, and 3200–3600 lux light intensity. The statistical differences in the number of adults between the two choices were analyzed using the chi-square test.

## 5. Conclusions

In summary, this study has identified the functions of the AspiOBP1 and AspiOBP2 proteins in binding and transporting host volatiles. AspiOBP1 is likely involved in this mechanism, with ARG-79 potentially serving as a critical amino acid site for this binding process. AspiOBP2 appears to play a supportive role in this mechanism. This research lays the groundwork for a deeper exploration of the host selection mechanisms of *A. spiniferus* and provides theoretical support for the development of novel compounds and RNA-based pesticides aimed at controlling this pest.

## Figures and Tables

**Figure 1 ijms-26-08784-f001:**
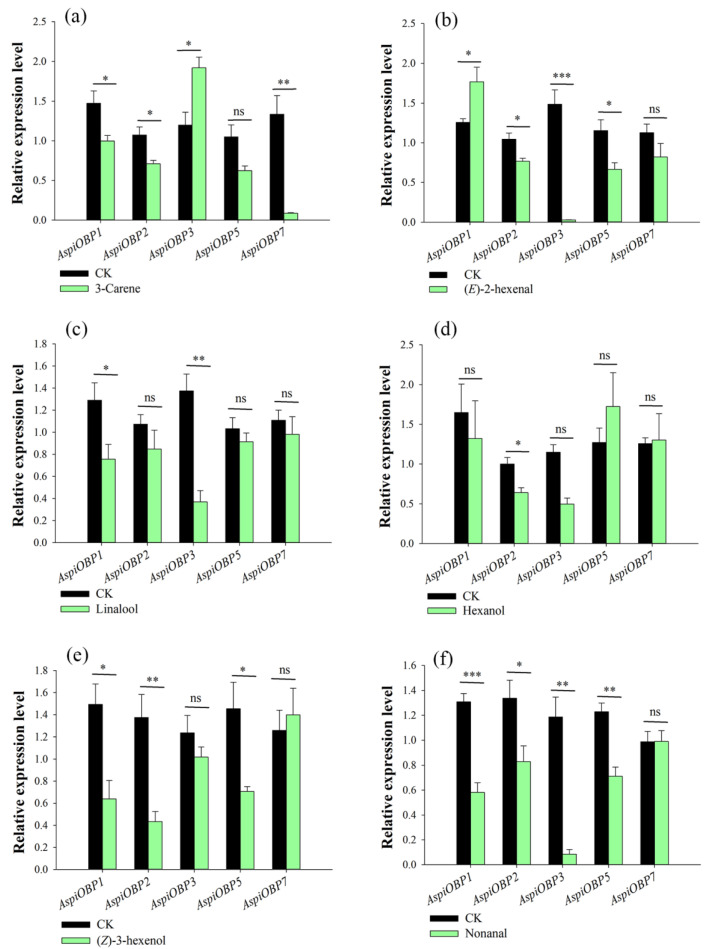
Effect of exposure to six host volatiles on transcript levels of *AspiOBP* genes. Templates from *A. spiniferus* treated with four attractants [3-carene (**a**), (*E*)-2-hexenal (**b**), linalool (**c**), hexanol (**d**)] and two repellents [nonanal (**f**), (*Z*)-3-hexenol (**e**)] were compared with those from untreated *A. spiniferus*. Bars indicate standard errors. All experiments were done in four replicates, using a two-tailed Student’s *t* test. * *p* < 0.05; ** *p* < 0.01; *** *p* < 0.001; ns indicate non-significant difference.

**Figure 2 ijms-26-08784-f002:**
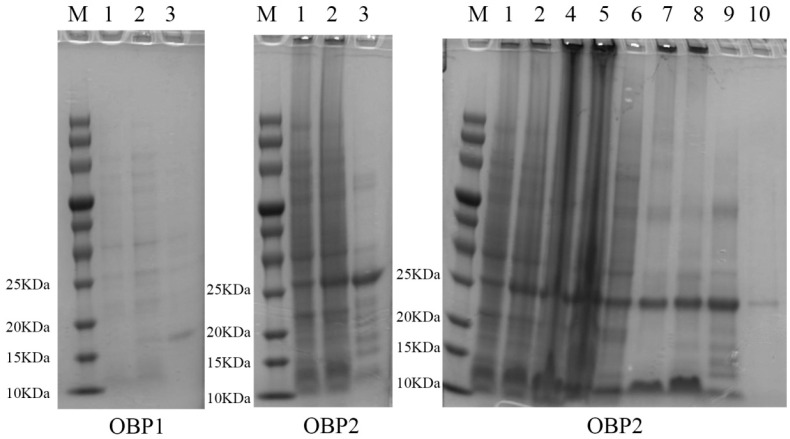
Bacterially expressed AspiOBP1 and AspiOBP2. Label 1, non-induced with IPTG; 2, induced with IPTG; 3, protein; 4, supernatant; 5, insoluble aggregate; 6, protein eluted from 20 mmol/L imidazole; 7, protein eluted from 50 mmol/L imidazole; 8, protein eluted from 100 mmol/L imidazole; 9, protein eluted from 250 mmol/L imidazole; 10, protein eluted from 500 mmol/L imidazole; M, molecular weight marker.

**Figure 3 ijms-26-08784-f003:**
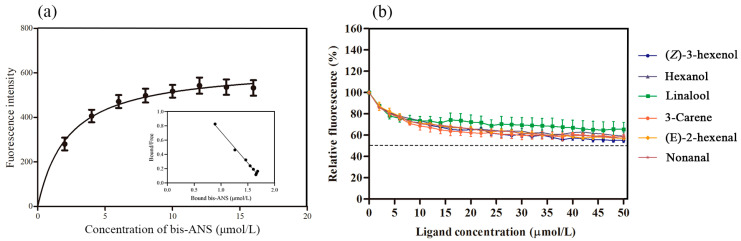
Ligand-binding assays of recombinant AspiOBP2. (**a**): Binding curve and Scatchard formula analysis of bis-ANS with AspiOBP2. (**b**): Binding curves of AspiOBP2 with six ligands.

**Figure 4 ijms-26-08784-f004:**
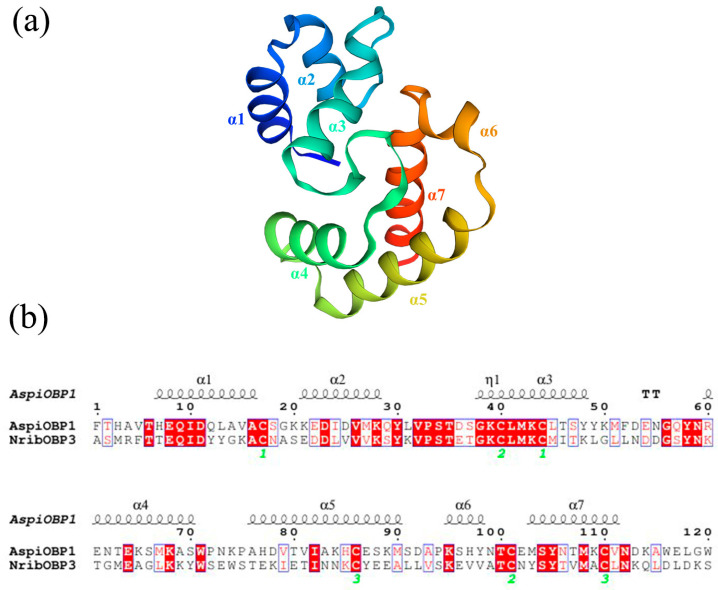
Sequence and structure of AspiOBP1. (**a**): Three-dimensional structure of AspiOBP1; (**b**): Alignment of amino acid sequences between AspiOBP1 and template.

**Figure 5 ijms-26-08784-f005:**
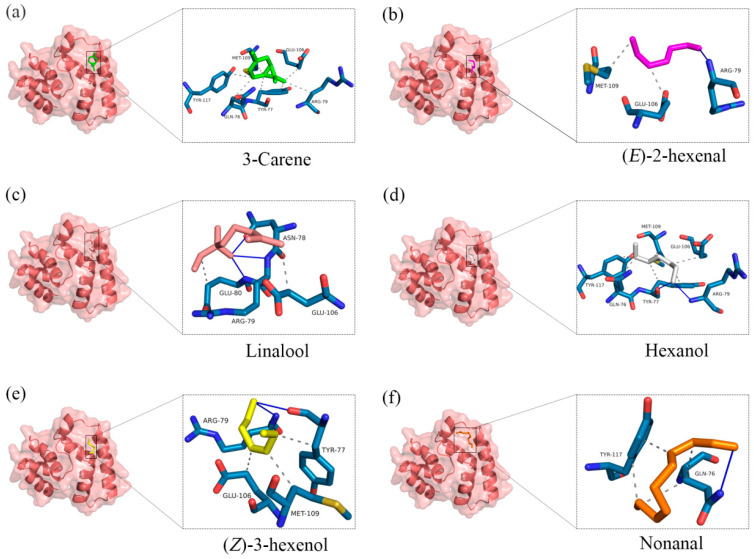
The molecular docking results of AspiOBP1 to (**a**): 3-carene, (**b**): (*E*)-2-hexenal (**c**): linalool, (**d**): hexanol, (**e**): (*Z*)-3-hexenol, (**f**): nonanal. The first and third columns show the optimal binding mode of AspiOBP1 to ligands. The second and fourth columns show the three-dimensional diagram of docking results. Blue solid lines show the hydrogen bonds, black dashed lines show the hydrophobic interactions.

**Figure 6 ijms-26-08784-f006:**
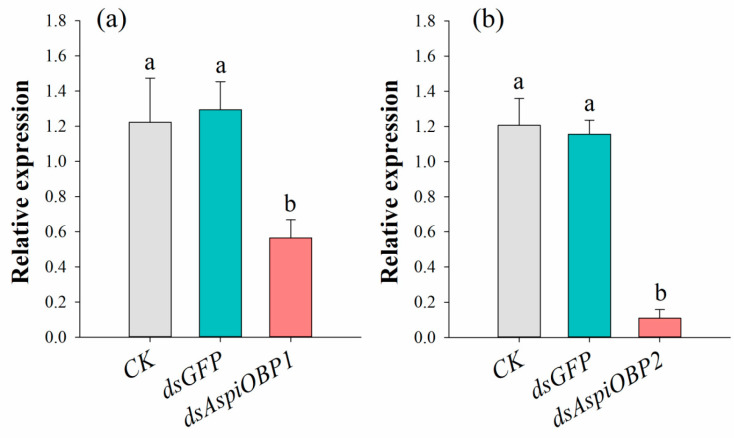
Effect of RNAi treatment on the transcript levels of *AspiOBP1* (**a**) and *AspiOBP2* (**b**). Different letters above bars indicate statistically significant differences based on one-way analysis of variance using the Tukey’s-b multiple range test (*p* < 0.05).

**Figure 7 ijms-26-08784-f007:**
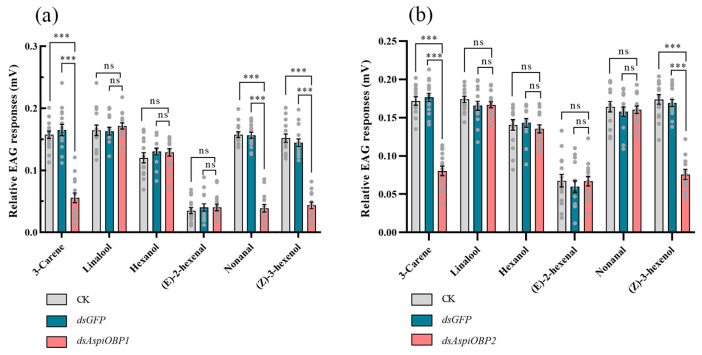
Effect of silencing *AspiOBP1* (**a**) and *AspiOBP2* (**b**) on EAG recordings of *A. spiniferus* to host volatiles. Data are shown as mean ± SEM, *** *p* < 0.001, one-way analysis of variance, Tukey’s-b multiple range test. ns indicate non-significant difference.

**Figure 8 ijms-26-08784-f008:**
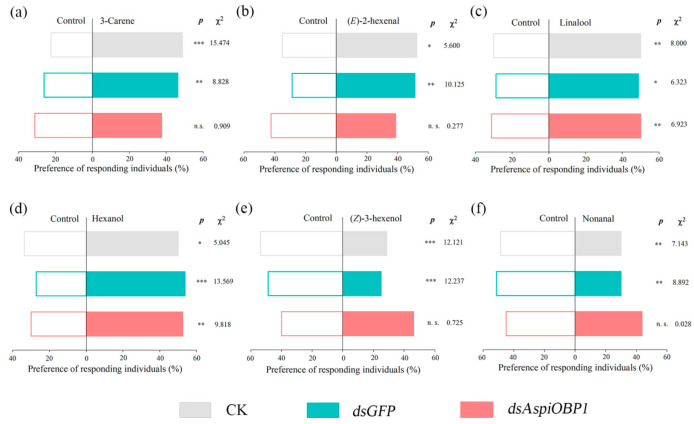
Behavioral responses of CK, *dsGFP*-feeding and *dsAspiOBP1*-feeding *A. spiniferus* to 3-carene (**a**), (*E*)-2-hexenal (**b**), linalool (**c**), hexanol (**d**), (*Z*)-3-hexenol (**e**), nonanal (**f**). Asterisks indicate significant difference (* *p* < 0.05, ** *p* < 0.01, *** *p* < 0.001) preference between control and odor sources via a Chi-square test. The number to the right of the asterisk is the Chi-square. n.s. means no significant difference (*p* > 0.05).

**Figure 9 ijms-26-08784-f009:**
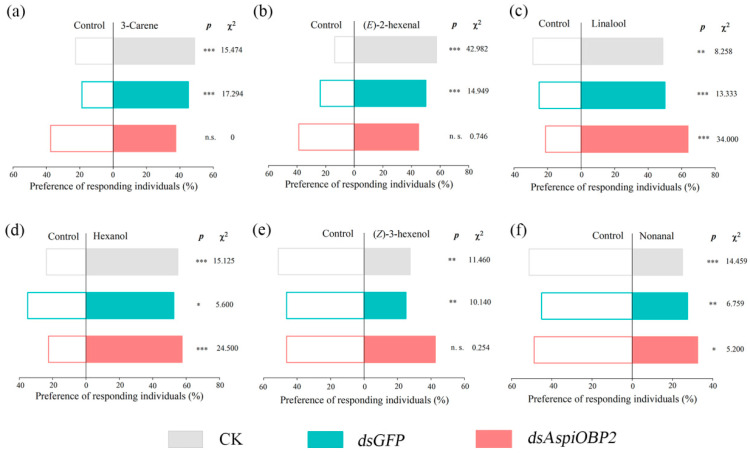
Behavioral responses of CK, *dsGFP*-feeding and *dsAspiOBP2*-feeding *A. spiniferus* to 3-carene (**a**), (*E*)-2-hexenal (**b**), linalool (**c**), hexanol (**d**), (*Z*)-3-hexenol (**e**), nonanal (**f**). Asterisks indicate significant difference (* *p* < 0.05, ** *p* < 0.01, *** *p* < 0.001) preference between control and odor sources via a Chi-square test. The number to the right of the asterisk is the Chi-square. n.s. means no significant difference (*p* > 0.05).

**Table 1 ijms-26-08784-t001:** Binding energy of AspiOBP1 to ligands.

Proteins	Ligands	Docking Energy (kcal/mol)
AspiOBP1	3-Carene	−3.07
	(*E*)-2-Hexenal	−2.47
	(*Z*)-3-Hexenol	−3.10
	Linalool	−1.78
	Hexanol	−1.99
	Nonanal	−1.89

## Data Availability

The data that support the findings of this study are available from the corresponding author upon reasonable request.

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
