# Peer review of "The Odorant-Binding Proteins AspiOBP1 and AspiOBP2 in Aleurocanthus spiniferus Are Involved in the Perception of Host Volatiles"

_ijms, 2025, doi:10.3390/ijms26188784_

Round 1

Reviewer 1 Report

Comments and Suggestions for Authors

The citrus spiny whitefly, Aleurocanthus spiniferus (Quaintance), is considered one of the major threats to citrus production in Asia, Australasia, and the Nearctic zone. Aleurocanthus spiniferus has several hosts of botanical families except for Citrus spp. and Theaceae. It is still in the process of adapting to new hosts. Odorant-binding proteins (OBPs) are capable of recognizing a variety of volatile compounds and act as chemosensory solubilizers, transporters, and ligand filters, mediating the activation of odorant receptors. The diversity and non-homology of OBPs can serve as an advantage for developing species-specific semiochemicals and even insecticides. Recently, five OBPs in A. spiniferus have been identified through transcriptomic analysis. (E)-2-hexenal, linalool, 3-carene, and hexanol have been confirmed to act as attractants for A. spiniferus, while nonanal and (Z)-3-hexenol function as repellents. However, the mechanism of AspiOBPs how to bind and transport host volatiles remains still unclear.

In this study, the authors explored the functions of key OBPs mediating host selection by measuring the in vivo and in vitro binding capabilities of OBPs from A. spiniferus to host volatiles. They found that AspiOBP1 is likely involved in binding and transporting host volatiles, with ARG-79 potentially serving as a critical amino acid site for this binding process. AspiOBP2 appears to play a supportive role in transporting host volatiles. Their findings not only elucidate the binding mechanisms between OBPs of A. spiniferus and host volatiles, but also provide new targets for the development of novel plant-derived insecticides and RNA-based pesticides to control this pest in future. However, in the present manuscript, some issues listed as major concerns and minor conerns need to be addressed as follow:

Major Concerns:

  1. In the section of Introduction, the authors should point out the shortcoming of previous rsearch in the second to last paragraph and provide the significance of this study in practical applications in the last paragraph.
  2. According to the result of Figure 1, most of the transcript levels of AspiOBP1, 2, 3  were significantly changed afer exposure to four attractants [3-carene, (E)-2-hexenal, linalool, hexanol] and two repellents [nonanal, (Z)-3-hexenol]. I don’t know why the authors did not explore the funcation of AspiOBP3 gene in the subsequent experiments? In addition, the authors should explain why no significant difference was observed in the relative expression level of AspiOBP3, but significant difference in the the relative expression level of AspiOBP2 in the Figure 1 d?
  3. In “7. Synthesis of Double-Stranded RNA (dsRNA)”of Materials and Methods, the titile should be revised as “4.7. Synthesis of Double-Stranded RNA (dsRNA) and Treatment”. Additionally, a schematic diagram of the plastic cup for dsRNA treatment in Lines 380-386 should be provided to easily understand. Last, I recommend the authors can directly explore the interaction of AspiOBP1 and AspiOBP2 using RNAi experiment with a mixture of dsAspiOBP1 and dsAspiOBP2.
  4. In the present manuscript, there are a lot of abbreviations. So, I recommend the authors should add a list of abbrevations.
  5. In References including 9, 10, 15, 18-21, 25, 26, 28-35, 38-41, 43-46, 50, 53-56, and 58-66, the Latin scientific names of species should be italicized.

Minor concerns:

  1. A space should be inserted between theunit of temperature or others and Arabic numerical values.
  2. In Lines 16-17, the sentence of “Under exposure to host volatiles, the two OBPs, AspiOBP1 and AspiOBP2, exhibited significant differential transcriptional regulation.”is not comprehensive and accurate enough because AspiOBP3 also exhibited significant differential transcriptional regulation according to Figure 1.  
  3. In Lines 25-26, “whereas AspiOBP2 likely serves a complementary role in this process.” is not accurate because “AspiOBP2 exhibited no binding to any of the six tested volatiles in fluorescent competitive binding assays”. So, at least AspiOBP2 does not play a complementary role in the binding of host volatiles”.
  4. In Line 102, “*P<0.05; **P<0.01; ***P<0.001”should be revised as “*P < 05; **P < 0.01; ***P < 0.001”.
  5. In Lines 168-169, “. P< 0.05, one-way analysis of variance, Tukey’s-b multiple range test” should be revised as “based on one-way analysis of variance using the Tukey’s-b multiple range test (P < 0.05) ”.
  6. In Line 393, “dsOBP1, dsOBP2”should be corrected as “dsAspiOBP1, dsAspiOBP2”.

Finally, I hope the authors can use these to correct the same problem for the rest.

Author Response

Responses to comments

Comment_01

In the section of Introduction, the authors should point out the shortcoming of previous research in the second to last paragraph and provide the significance of this study in practical applications in the last paragraph.

We are grateful for your constructive feedback. As suggested, we have revised the Introduction to clearly address the limitations of previous studies in the second-to-last paragraph and emphasize the practical significance of our work in the last paragraph. These modifications enhance the clarity and relevance of our manuscript. We sincerely appreciate your time and effort to improve the quality of our paper. (Line 79-81, 86-89)

Comment_02

According to the result of Figure 1, most of the transcript levels of AspiOBP1, 2, 3 were significantly changed after exposure to four attractants [3-carene, (E)-2-hexenal, linalool, hexanol] and two repellents [nonanal, (Z)-3-hexenol]. I don’t know why the authors did not explore the funcation of AspiOBP3 gene in the subsequent experiments?

We sincerely thank you for this insightful comment and for highlighting the interesting expression pattern of AspiOBP3. In our experimental design, we focused on AspiOBP1 and AspiOBP2 for functional characterization because these two genes exhibited significant transcriptional changes in response to more than five different host volatiles. This broad-spectrum responsiveness suggests that AspiOBP1 and AspiOBP2 may play more generalized roles in chemoreception and host volatile detection. Although AspiOBP3 also showed significant changes under certain treatments, its responses were comparatively less consistent across a wider range of ligands. Therefore, we prioritized the two OBPs with the broadest response profiles for further functional analysis to better understand their potential roles in host recognition. We agree that further study of AspiOBP3 could be valuable and will consider this in our future research.

Comment_03

In addition, the authors should explain why no significant difference was observed in the relative expression level of AspiOBP3, but significant difference in the relative expression level of AspiOBP2 in the Figure 1 d?

We sincerely appreciate your important comments regarding the differential expression patterns of AspiOBP2 and AspiOBP3 in Figure 1d. All expression analyses in this study strictly followed standardized experimental protocols, including three biological replicates and three technical replicates, and used validated reference genes for normalization to ensure result reliability. Comparisons between the control (CK) and treatment groups were performed using a two-tailed Student’s t-test, with complete detailed results provided in Table A1. The absence of a significant difference in AspiOBP3 expression, as opposed to AspiOBP2, may be attributed to several factors, such as higher biological variability among samples, lower sensitivity of this gene to the specific compounds, or inherent differences in the regulatory mechanisms of the two OBPs. Furthermore, data normality and homogeneity of variance were rigorously verified before analysis. We sincerely thank you for your insightful questions, which have greatly enhanced the analytical rigor and clarity of the manuscript.

Comment_04

In “7. Synthesis of Double-Stranded RNA (dsRNA)” of Materials and Methods, the title should be revised as “4.7. Synthesis of Double-Stranded RNA (dsRNA) and Treatment”.

Thank you for your meticulous attention to the details of the manuscript. Your suggestions have been incorporated into the latest revised version. (Line 402)

Comment_05

Additionally, a schematic diagram of the plastic cup for dsRNA treatment in Lines 380-386 should be provided to easily understand.

Thank you very much for your insightful comments and valuable suggestions. We greatly appreciate the time and effort you have dedicated to reviewing our manuscript. Regarding your suggestion about providing a schematic diagram of the plastic cup device used for dsRNA treatment, we would like to mention that a detailed illustration of this setup has already been published in our previous paper entitled “Two chemosensory proteins in Aleurocanthus spiniferus are involved in the recognition of host VOCs” (2024). For clarity and convenience, we have cited this reference in the manuscript (Lines 403-404). Should it be necessary, we would be pleased to provide the schematic diagram separately as supplementary material. We sincerely thank you again for your thorough review and constructive feedback.

Comment_06

Last, I recommend the authors can directly explore the interaction of AspiOBP1 and AspiOBP2 using RNAi experiment with a mixture of dsAspiOBP1 and dsAspiOBP2.

Thank you very much for taking the time to provide such insightful and valuable suggestions. We highly appreciate your positive feedback, especially your thoughtful recommendation regarding the use of RNA interference to explore the functional interaction between AspiOBP1 and AspiOBP2. We fully agree that employing a mixed dsRNA approach (dsAspiOBP1 + dsAspiOBP2) would offer a more direct and comprehensive strategy to investigate potential synergistic or compensatory roles between these two proteins.

At this stage of our research, the initial experimental design focused primarily on elucidating the individual physiological functions of AspiOBP1 and AspiOBP2 to establish a clear functional baseline. As a result, the mixed RNAi approach was not included in the current study. We recognize that your suggestion is crucial for gaining a deeper understanding of the collaborative mechanisms between these two proteins. This is an excellent recommendation, and we will incorporate this strategy into our future research plans. We believe that such experiments will provide more profound functional insights and robust evidence regarding their interaction. Accordingly, we will design and conduct the corresponding RNAi experiments in subsequent studies to address this important question. Thank you once again for your constructive comments, which have undoubtedly significantly enhanced the quality and depth of our work.

Comment_07

In the present manuscript, there are a lot of abbreviations. So, I recommend the authors should add a list of abbreviations.

Thank you for your meticulous attention to the details of the manuscript. Your suggestions have been incorporated into the latest revised version. (Line 33)

Comment_08

5.In References including 9, 10, 15, 18-21, 25, 26, 28-35, 38-41, 43-46, 50, 53-56, and 58-66, the Latin scientific names of species should be italicized.

Thank you for your meticulous attention to the details of the manuscript. Your suggestions have been incorporated into the latest revised version.

Comment_09

A space should be inserted between the unit of temperature or others and Arabic numerical values.

Thank you for your meticulous attention to the details of the manuscript. Your suggestions have been incorporated into the latest revised version.

Comment_10

In Lines 16-17, the sentence of “Under exposure to host volatiles, the two OBPs, AspiOBP1 and AspiOBP2, exhibited significant differential transcriptional regulation.” is not comprehensive and accurate enough because AspiOBP3 also exhibited significant differential transcriptional regulation according to Figure 1.

Thank you for your meticulous attention to the details of the manuscript. Your suggestions have been incorporated into the latest revised version. (Line 16)

Comment_11

In Lines 25-26, “whereas AspiOBP2 likely serves a complementary role in this process.” is not accurate because “AspiOBP2 exhibited no binding to any of the six tested volatiles in fluorescent competitive binding assays”. So, at least AspiOBP2 does not play a complementary role in the binding of host volatiles”.

We appreciate your valuable input. Acknowledging that the statement in question was poorly phrased, it has been removed without compromising the integrity of the content.

Comment_12

4.In Line 102, “*P<0.05; **P<0.01; ***P<0.001”should be revised as “*P < 0.05; **P < 0.01; ***P < 0.001”.

Thank you for your meticulous attention to the details of the manuscript. Your suggestions have been incorporated into the latest revised version. (Line 110)

Comment_13

In Lines 168-169, “. P< 0.05, one-way analysis of variance, Tukey’s-b multiple range test” should be revised as “based on one-way analysis of variance using the Tukey’s-b multiple range test (P < 0.05) ”.

Thank you for your meticulous attention to the details of the manuscript. Your suggestions have been incorporated into the latest revised version. (Line 177-178)

Comment_14

In Line 393, “dsOBP1, dsOBP2” should be corrected as “dsAspiOBP1, dsAspiOBP2”

Thank you for your meticulous attention to the details of the manuscript. Your suggestions have been incorporated into the latest revised version. (Line 421)

Reviewer 2 Report

Comments and Suggestions for Authors

Dear Authors,

I read your paper with great interest. The topic of OBPs and their potential use is an interesting alternative to the currently used methods. However, I have a few questions and comments I'd like to have answered.

Line 72 - Why did you decide to test these particular compounds?

Point 2.2 - I'm missing an explanation, or perhaps I didn't catch it, why only OBP1 and OBP2 were tested in this section, and the other three OBPs weren't included?

Point 4.1, line 259 - Acclimation issues should be described in more detail - how long did it last? What was the mortality rate during this time?

Line 288 - SRA is Sequence Read Archive, not Short Read Archive

Line 296 - This purity and integrity analysis method is definitely outdated. It would be worthwhile to at least use Tapestation or another spectrophotometric system that allows for the assessment of A260/A230 and A260/A280 ratios.

Line 300 - "Taq Plus Master Mix" - add the manufacturer's name

Line 305 - Does this mean that the PCR product itself was not sequenced and compared against databases?

Line 308 - "BamHI and EcoRI" - add the manufacturer's name

Line 316 - I don't understand how these cultures were selected. How was the amino acid sequence analyzed? Was MS performed?

Lines 318-320 - The first sentence states that the plasmids were extracted, but the next sentence repeats this. I also understand that the nucleotide sequence was analyzed, right? For example, by sequencing the product.

Line 323 - Why was the expression level low? And why wasn't it repeated under different conditions until the correct expression level was achieved? Perhaps it was worth changing the expression system?

Line 332 - How were the dialysis and ultrafiltration performed?

Line 334 - Why weren't the obtained proteins verified by mass spectrometry? Were such unverified proteins tested?

Round 2

Reviewer 1 Report

Comments and Suggestions for Authors

The authors have responded to my comments point by point. I’m satisfied with most revisions they have made. However, one Major Concern and one Minor Concern still need to be addressed as currently written in the revised manuscript. So I think it is suitable for publication after Minor Revision.

Major Concern

1. In References including 9, 10, 15, 18-21, 25, 26, 28-35, 38-41, 43-46, 50, 53-56, and 58-66, the Latin scientific names of species should be italicized.

Although the authors have claimed to address this issue, but the revised manuscript still reamains unchanged.

Minor Concerns

1. A space should be inserted between theunit of temperature or others and Arabic numerical values.

Although the authors have claimed to address this issue, but the revised manuscript still reamains unchanged.

Author Response

We sincerely apologize that some of the revisions to the manuscript did not meet your expectations. Due to an oversight on our part, the Latin names in the references were not italicized after being copied and pasted. We have now thoroughly addressed this issue in the latest version to prevent it from happening again. Furthermore, the absence of spaces between the numbers and the symbols ℃ and % was an error on our part, attributed to variations in usage conventions across different countries. This has also been corrected in the newly uploaded manuscript. Thank you for your meticulous attention to the details of the article, which is immensely important for our paper.